# Real-Time Cucumber Target Recognition in Greenhouse Environments Using Color Segmentation and Shape Matching

**Wenbo Liu \*, Haonan Sun, Yu Xia and Jie Kang**

School of Electrical and Control Engineering, Shaanxi University of Science and Technology, Xi'an 710021, China; 210612072@sust.edu.cn (H.S.); yuxia@sust.edu.cn (Y.X.); kangjie@sust.edu.cn (J.K.)
\* Correspondence: wbliu@sust.edu.cn

**Abstract:** Accurate identification of fruits in greenhouse environments is an essential need for the precise functioning of agricultural robots. This study presents a solution to the problem of distinguishing cucumber fruits from their stems and leaves, which often have similar colors in their natural environment. The proposed algorithm for cucumber fruit identification relies on color segmentation and form matching. First, we get the boundary details from the acquired image of the cucumber sample. The edge information is described and reconstructed by utilizing a shape descriptor known as the Fourier descriptor in order to acquire a matching template image. Subsequently, we generate a multi-scale template by amalgamating computational and real-world data. The target image is subjected to color conditioning in order to enhance the segmenacktation of the target region inside the HSV color space. Then, the segmented target region is compared to the multi-scale template based on its shape. The method of color segmentation decreases the presence of unwanted information in the target image, hence improving the effectiveness of shape matching. An analysis was performed on a set of 200 cucumber photos that were obtained from the field. The findings indicate that the method presented in this study surpasses conventional recognition algorithms in terms of accuracy and efficiency, with a recognition rate of up to 86%. Moreover, the system has exceptional proficiency in identifying cucumber targets within greenhouses. This attribute renders it a great resource for offering technical assistance to agricultural robots that operate with accuracy.

**Keywords:** greenhouse environments; fruit recognition; color split; multi-scale; shape match

## 1. Introduction

Cucumber, which is among the top ten vegetables worldwide, is a highly significant and commercially advantageous crop. Cucumbers are highly nutritious, including significant amounts of vitamin B2, vitamin C, carotene, and other beneficial compounds. They are known for their beauty and skincare benefits [1]. Based on data from the Food and Agriculture Organization of the United Nations (FAO), the global cucumber production area in 2020 was 2,261,000 hectares, with a total production of 91,258,000 tonnes. China continually holds the top position globally in terms of the cucumber cultivation area and yield, making it a crucial industry for adjusting the industrial structure, generating farmer income, and promoting rural economic growth. Nevertheless, the extensive cultivation area and high productivity result in labor expenses comprising more than 50% of the overall production costs, emphasizing the practical importance of investigating precise operations of agricultural robots [2,3]. The precise and swift identification of fruit targets is a fundamental focus of agricultural robotics research. However, the intricate nature of the natural development environment, difficulties with near-color backgrounds, occlusion concerns, and variations in illumination conditions provide significant challenges in recognizing fruit targets [4].

The rise of information technology has led to a growing utilization of computer vision for intricate target identification. Extracting fruit color is a relatively simple task as long as

there is a noticeable difference in color between the fruit being targeted and its background. In specific circumstances, color characteristics are employed to identify fruits when the colors of the fruit and the background are alike, as seen with green apples, unripe green tomatoes, and cucumbers. Vitzrabin [5] initially partitioned the target image into roughly equal rectangular sub-images. They computed the thresholds for each color component of the RGB image to achieve a uniform region of illumination. Subsequently, they transformed it into a three-dimensional spatial image using the natural difference index (NDI) and determined three distinct thresholds for each NDI dimension in every sub-image. This approach enabled the detection of red bell peppers with a high success rate, even in situations with significant variations in illumination. Moghimi [6] successfully segmented green peppers by merging the H1 and S1 components of the HSI spatial model in the smoothing region, along with the ultra-green operator EXG (EXG = 2G-R-B). Zhang [7] constructed the H1 component of the super green operator EXG, the normalized g component, and the S1 component in both the RGB and HSV color spaces. The integration of a support vector machine and threshold classification enabled them to attain recognition of green apples. Li [8] substituted the green component in the RGB color space with the hue component (H2) in the HSV color space to create the composite image (RBH). This modification increased the distinction between the green tomato and its surrounding environment. The researchers employed a thresholding segmentation method to accomplish the initial segmentation of the green tomato.

The color of fruits in natural settings can be readily influenced by various lighting conditions, resulting in color deficit. On the other hand, form features, which are another important characteristic of fruits, can be used to accurately separate and identify fruits. However, it is more difficult to describe and extract these features. Researchers have used shape descriptors, such as the histogram of oriented gradient (HOG), deformable part model (DPM), and Fourier descriptors, in target recognition systems. Bao [9] performed an examination of the morphology and developmental stage of cucumbers, which resulted in the creation of a repository of templates encompassing various dimensions and bending angles. Subsequently, the team utilized a template matching technique to accurately identify cucumbers.

To enhance the amount of image data, researchers have introduced hyperspectral cameras, infrared cameras, and thermal imagers for image acquisition [10]. Okamoto [11] utilized a hyperspectral camera with 60 spectral bands ranging from 369 nm to 1042 nm to capture images of green citrus. Through spectral analysis, they selected specific bands with distinctive features. They then applied a thresholding technique to remove bright and dark backgrounds in the images. Finally, by integrating all the bands, they established a linear discriminant to accurately segment green leaves and green citrus. Wendel [12] employed a handheld hyperspectral camera to capture images of mangoes in an orchard. They used two different methods, namely, PLS (partial least square) and simple CNN (convolutional neural network), for mango recognition and ripeness estimation, respectively. Despite the ability of the hyperspectral camera to capture extensive color data, its operation is intricate, the settings it operates in are challenging, and the process of acquiring images is highly demanding. The conditions are demanding, and the procedure of acquiring the photograph is time-consuming.

For fruit recognition, three-dimensional picture features are employed as supplementary attributes. Both Tian [13] and Liu [14] have exploited depth photographs of fruits and vegetables to aid in fruit recognition. The fruit's location and center point were identified by rotating the center of the vortex's gradient vector of the depth image, which was mapped to a 2D space. They subsequently used shape fitting to successfully recognize targets, with a particular focus on round fruits like oranges and apples. Barnea [15] incorporated the detection of bell peppers to identify various 3D image features, including 3D surface normal features, 3D plane symmetry features, and elliptical surface highlight features. Kusumam [16] employed a Kinect V2 camera to capture RGBD images of ripe broccoli in the field. They then utilized the VFH (viewpoint feature histogram) feature operator to

describe the target and employed an SVM classifier to classify the feature vectors and locate the head of the ripe broccoli. Nyarko [17] introduced the CTI (convex template instance) operator to represent the 3D shape of fruits for the purpose of identifying convex fruits like tomatoes and apples. The KNN (K-nearest neighbor) feature operator was employed to describe the fruits, and then, the SVM classifier was used to classify the feature vectors, enabling the detection and localization of ripe broccoli heads.

Typically, fruit recognition research primarily concentrates on a single category of picture characteristics to identify the desired object. Nevertheless, many studies have employed more extensive techniques to apply specific visual characteristics to fruit identification. Tao [18] introduced an algorithm for extracting texture features based entirely on color-related local binary patterns. Through the integration of color and texture characteristics and the utilization of the nearest neighbor classifier, the researchers successfully accomplished fruit and vegetable categorization, resulting in a minimum 5% improvement in recognition accuracy.

Promising results have been achieved in the application of deep learning technology for fruit recognition in complicated situations [19–22]. Gené-Mola [23] utilized a Kinect camera to capture fruit images, employing RGB images, depth images, and reflected signal intensity maps. These images were merged into a 5-channel image and used as input for the Faster RCNN algorithm to identify Red Fuji apples. Koirala [24] improved the YOLO V3 backbone network by combining the characteristics of the compact YOLO V2 network with its few layers and high speed and the accurate residual structure of the original YOLO V3 network. This modification resulted in enhanced detection speed and accuracy.

Current agricultural mobile robots require uncomplicated algorithms, minimal energy usage, affordability, and a user-friendly operation to enhance their utilization. They must also ensure precise and efficient identification of fruit and vegetable targets [25]. Deep learning algorithms outperform traditional vision algorithms in recognizing targets in complex environments, but they need extensive data for training. Insufficient data limits their performance. Moreover, training with a large dataset consumes more energy and demands significant computational resources and time, leading to high hardware requirements and increased costs. Deep learning models are often considered "black boxes" due to their complex internal workings and decision-making processes. Additionally, these models may overfit the training data, leading to subpar performance on new or unseen data, and are susceptible to noise or minor variations in the input data.

This paper employs conventional vision algorithms for target recognition based on color and shape features. These algorithms are straightforward to implement and have minimal computational and feature extraction requirements, low hardware costs, high interpretability, and good recognition accuracy with limited data. Additionally, they are energy-efficient, making them well suited for agricultural mobile robots.

## 2. System Architecture

This study integrates the color characteristics and shape characteristics of the target in order to achieve target recognition. The method follows the precise steps outlined below:

Initially, we employ an industrial camera to gather target photos directly at the greenhouse location and select a substantial quantity of cucumber fruit samples. In order to simplify the process of extracting cucumber form characteristics, the cucumber samples are photographed on white A4 paper to acquire sample photos.

Next, the image data undergo preprocessing. This involves applying image preprocessing techniques to all sample images to extract shape features. Normalized Fourier shape descriptors are then used to perform shape description and reconstruction. This process eliminates shape information that exhibits significant changes in shape features. The result is the template contour, which represents the standard shape of a cucumber fruit. First, the template image of a cucumber is obtained by computing multiple scales and angles. Next, color enhancement is applied to all target images. The target is then segmented using a threshold in the HSV color space, using color features to reduce interference. The

resulting segmented image is used to extract edge information using an edge detection algorithm. This edge information is then combined with the template image to perform shape matching using multiple templates.

The target recognition experiments are divided into two parts in the experimental design. First, target images are acquired using various traditional vision algorithms to conduct computer simulation experiments for multi-scale target recognition. The target recognition accuracy and real-time performance of different algorithms are compared, highlighting the superiority of the algorithm proposed in this paper. Subsequently, the proposed algorithm is implemented on hardware equipment for real-time target recognition experiments in greenhouse environments, further demonstrating its superiority and applicability. Figure 1 illustrates the precise procedure.

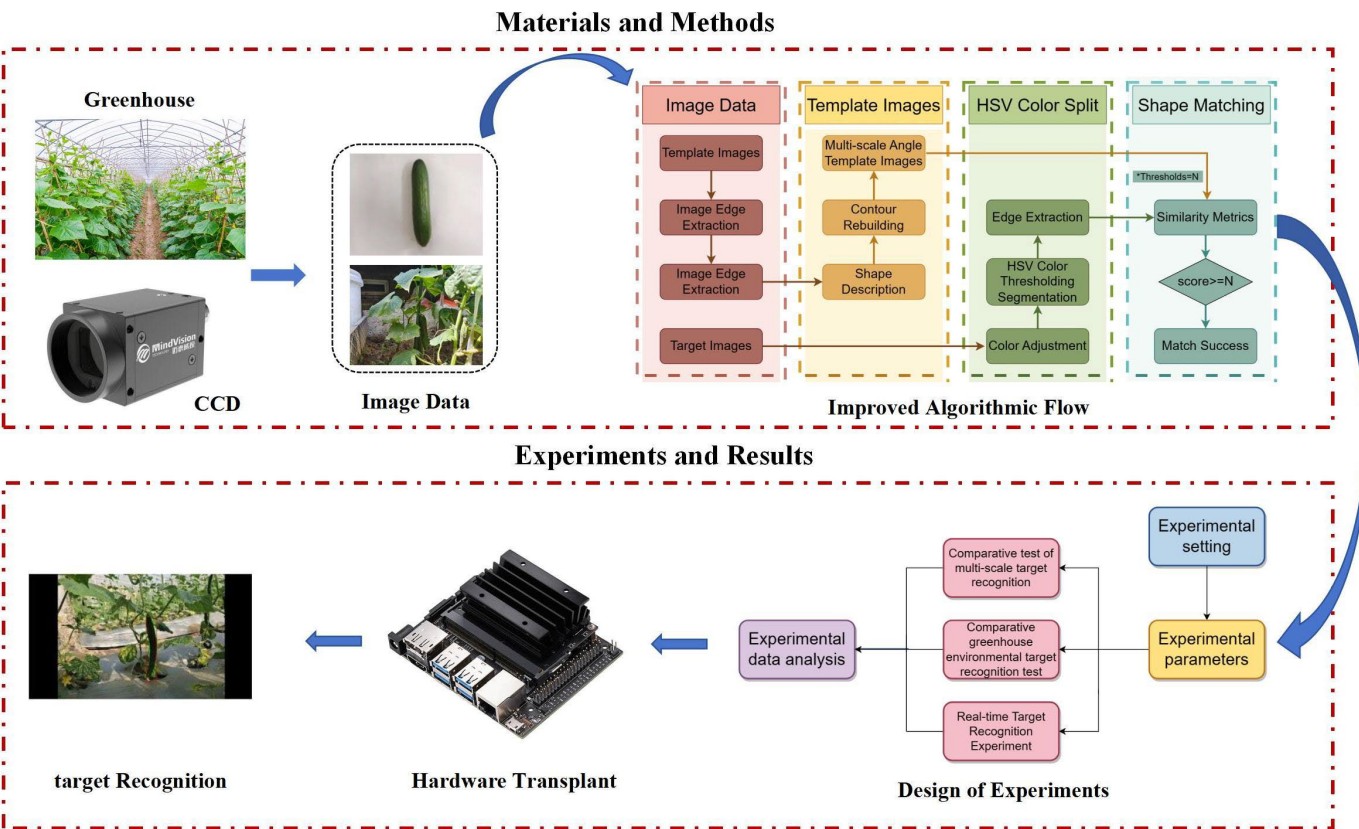

**Figure 1.** Comprehensive program. * Thresholds=N: The similarity matching score threshold is set to N.

## 3. Materials and Methods

### 3.1. Physical Components of a Computer System

3.1.1. Charge-Coupled Device (CCD) Camera

The camera utilizes a CCD or CMOS image sensor to capture a two-dimensional image of the target. This image sensor turns the photoelectric signal into an electrical signal, serving as the primary light-sensitive component. The MindVision industrial camera USB2.0 is utilized for capturing images. The particular parameters can be seen in Table 1.

**Table 1.** Cameras and lenses.

| Parametric | Configure |
|---|---|
| Name | MV-UBS130RC/MP-IR |
| Lens | 3.5~8 mm |
| Resolution | $1280 \times 960/1.4$ |
| Lens Interface | C/CS |
| CCD Size | $1/3''$ |

### 3.1.2. Photographic Optics

The lens focal length options for this market include 3.5 mm, 4 mm, 6 mm, 8 mm, 12 mm, and other specifications. Smaller focal lengths provide a wider range of movement for capturing good images. The formula for selecting a lens is derived from the imaging principle, as seen in Figure 2.

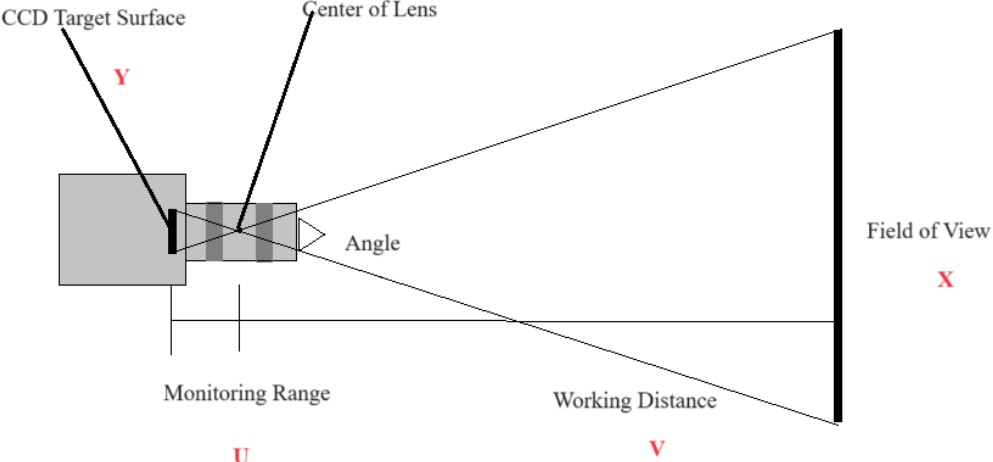

**Figure 2.** Foundational concepts of the camera image.

The lens selection formula can be derived from Figure 2. The ratio of X to CCD size is equal to the ratio of V to U.

Given that the cucumber's length is approximately 12–15 cm, the field of vision is set to 20 cm, the CCD size is 3.6 mm (1/3 type), the working distance is about 40 cm, and the focal length is calculated to be 7.2 mm; the lens parameter information is displayed in Table 1.

### 3.1.3. Additional Hardware

In order to minimize energy consumption and prolong the operational duration, a vision system that does not require a light source has been selected for cucumber-picking robots as the motors and background light sources consume a substantial amount of electricity.

The NVIDIA Jeston Nano 2GB series is chosen for the embedded development version due to its sufficient RAM capacity for the algorithm's moderate memory requirements.

The mobile robot uses a four-wheel independent drive for movement, multi-sensor fusion SLAM for tracking and navigation, an RTT real-time operating system for motion control, and environmental sensors to collect data in the greenhouse. It can also perform crop identification and positioning using a camera and target detection algorithm. Several hardware components are depicted in Figure 3.

### 3.2. Dataset

### 3.2.1. Example Image Data

For this investigation, a total of 200 cucumber targets were chosen as examples from greenhouse environments. The samples were photographed in Jingyang County, Xianyang City, Shaanxi Province, China. A CCD industrial camera with uniform illumination was used to capture the images on white A4 paper. There were no other interfering backgrounds in the images. The image size was 170 × 128 pixels, and they were 24-bit RGB maps. Figure 4 shows the image data of some of the samples.

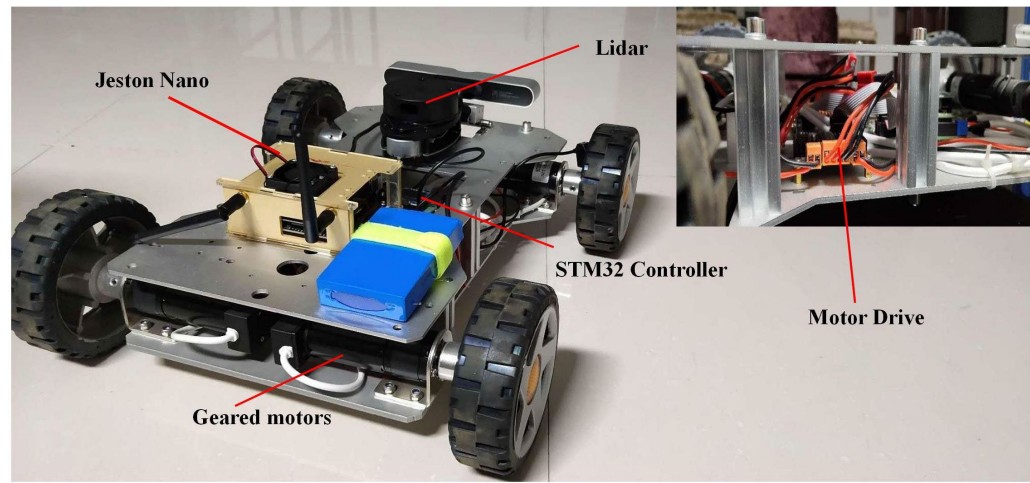

**Figure 3.** Mobile robotic system for agricultural applications.

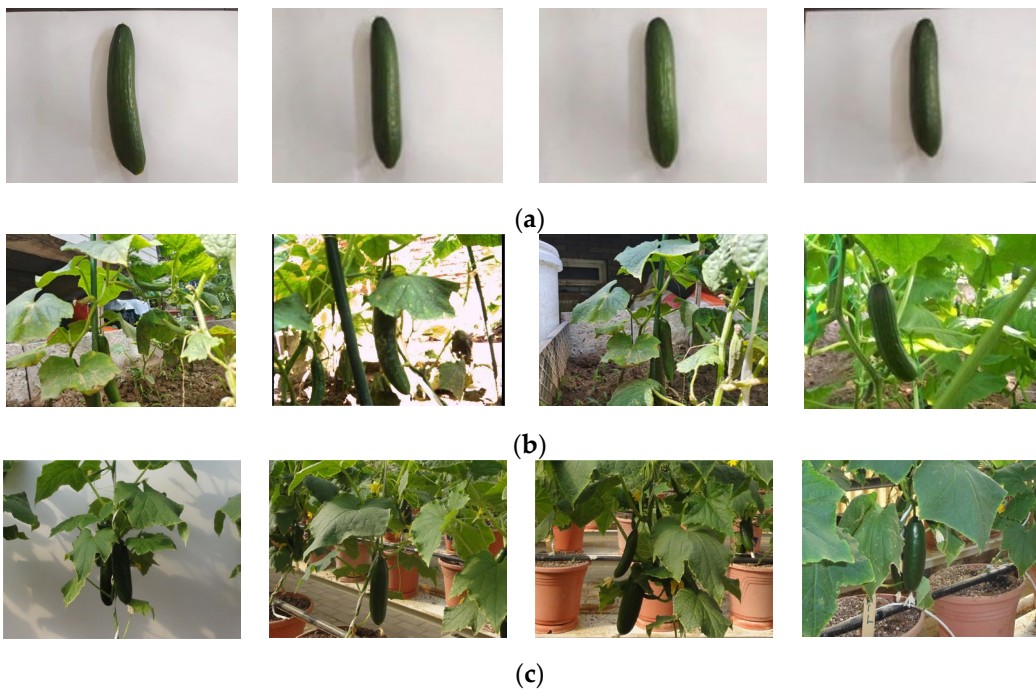

**Figure 4.** Image data: (**a**) sample images; (**b**) greenhouse shooting target images; (**c**) database images.

### 3.2.2. Target Image Data

In order to accurately represent the actual growth of cucumbers, we gathered 200 specific photographs of cucumbers growing naturally in a greenhouse under natural lighting conditions. The Kaggle dataset, the Cucumber Greenhouse Dataset, contains a total of 100 target photos. Several photos are depicted in Figure 4.

### 3.3. Geometric Characteristics

Cucumber fruits display a wide array of sizes and forms, requiring a fruit template that can accurately depict a vast range of fruit contour properties, even after undergoing changes in scale. In order to depict the form of the fruit target, we utilize a contour descriptor to accurately record and extract the outward characteristics of the fruit target [26,27].

3.3.1. Fourier Descriptor

Fourier descriptors are used as distinctive characteristics to create the outline of an image, thereby functioning as an image feature [28]. If we consider the goal shape profile as a closed curve, and a point $p(i)$ moves along this boundary curve with its complex coordinates $x(i) + jy(i)$, the period is determined by the perimeter of the closed curve. The given function can be represented mathematically as a Fourier series. Fourier descriptors refer to the coefficients $z(k)$ in the Fourier series that are defined as many coefficients. By choosing a coefficient term of $z(k)$ significant magnitude, the Fourier descriptor is capable of completely extracting contour details and reconstructing the contour of the object.

Let us assume that the boundary curve of the cucumber target is represented by a sequence of coordinates $s(k) = [x(k), y(k)]$. Each pair of coordinates is represented as a complex number, as shown in Equation (1).

$$s(k) = x(k) + iy(k), k = 0, 1, 2 \cdots K - 1 \tag{1}$$

In the context of the complex sequence $s(k)$, the horizontal x-axis corresponds to the real axis, while the vertical y-axis corresponds to the imaginary axis. Since the boundary curve of the cucumber target repeats itself every T units, the sequence of boundary coordinates can be regarded as a one-dimensional signal. The one-dimensional discrete Fourier transform (DFT) is defined according to Equation (2).

$$a(u) = \sum_{k=0}^{K-1} s(k)e^{-i2\pi uk/K}, \ u = 0, 1, 2, \cdots K - 1 \tag{2}$$

The coefficient $a(u)$ with complex values in Equation (2) is known as the Fourier descriptor of the border.

By applying an inverse Fourier transform to the complex coefficients $a(u)$, it is feasible to retrieve the original signal $s(k)$, specifically the coordinates of the boundary points of the target's set. By utilizing the initial P coefficients, Equation (3) demonstrates the approximation of the original boundary.

$$s(k) = \frac{1}{K}\sum_{u=0}^{P-1} a(u)e^{i2\pi uk/K}, \ k = 0, 1, 2 \cdots K - 1 \tag{3}$$

3.3.2. Fourier Descriptor in Normalized Form

To ensure rotational, translational, and scale-transformation invariance, it is necessary to normalize the descriptor of the target profile curve. This is because the initial point, size, and dimension of the curve have an impact on the size of the Fourier descriptor. Additionally, considering the scale change and rotational motion of the fruit target that needs to be recognized, normalization becomes crucial.

The process of normalization is carried out using the $a(1)$-basis to achieve the normalized Fourier descriptor, as stated in Equation (4).

$$D(u) = \frac{\|a(u)\|}{\|a(1)\|}, \ u = 0, 1, 2, \cdots K - 1 \tag{4}$$

Using the cucumber contour image as a reference, we performed scale modification and calculations to extract its first 8 normalized Fourier descriptors. The resulting data records may be found in Table 2.

**Table 2.** The initial eight normalized Fourier descriptors.

| Image Size | D(1) | D(2) | D(3) | D(4) | D(5) | D(6) | D(7) | D(8) |
|---|---|---|---|---|---|---|---|---|
| Original | 0.039948 | 0.067076 | 0.002810 | 0.022196 | 0.005083 | 0.011672 | 0.004651 | 0.004707 |
| Zoom in 1× | 0.037774 | 0.066517 | 0.003465 | 0.022031 | 0.005360 | 0.011737 | 0.004438 | 0.004625 |
| Zoom in 0.5× | 0.038786 | 0.070002 | 0.003502 | 0.023646 | 0.005702 | 0.013010 | 0.005489 | 0.005857 |
| Rotate 90° | 0.042617 | 0.062035 | 0.003128 | 0.018816 | 0.003987 | 0.009311 | 0.003844 | 0.002795 |

Based on the information presented in the table, it is evident that the normalized Fourier descriptor of the picture that has undergone scale transformation has minimal alteration. This indicates that the normalized Fourier descriptor $D(u)$ preserves its properties of being unaffected by rotation, translation, and scaling.

### 3.4. Calculation of Angles and Scales

Using only a single-scale template contour is insufficient for reliably identifying cucumber targets that vary significantly in size and orientation, based on observed development patterns in the actual world. Consequently, edge characteristics obtained from the reconstruction of standardized descriptors are adjusted in terms of rotation and scaling based on the transformation matrix in order to create templates with different sizes and orientations.

#### 3.4.1. Calculation of Angles

The size of the template profile rotation should consider both the angular range and the angular step size of the rotation. Optimizing the angular step size can decrease the number of templates and improve the accuracy of the matching process.

Let $D$ represent the distance from the center to the edge of the target cross in the template image. When the template picture is rotated by a certain angle $\Delta\omega$ measured in radians, the displacement of the edge point should not exceed half a pixel, as specified by the expression presented in Equation (5).

$$\sin\Delta\omega = \frac{1/2}{D} \tag{5}$$

When the value of $D$ is significantly larger than the value of $\Delta\omega$, the equation can be estimated using Equation (6).

$$\Delta\omega \approx \frac{1}{2D} \tag{6}$$

Given that the desired rotation has a time-based angular rang $[\pi, -\pi]$, the estimated number of template angle rotations can be calculated using Equation (7).

$$N_\varphi = \frac{2\pi}{\Delta\omega} \approx 4\pi D \tag{7}$$

#### 3.4.2. Calculation of Scale

The range for template picture zoom is defined by the target's real growth. The maximum distance from the central point of the template picture to any point on the edge is represented by the variable $D$. The displacement of points after a scale transformation is $\Delta s_e$. The offset of the edge point must not exceed half a pixel $\Delta s_e \leq 0.5$ as this would result in a mismatch issue. If we allow $\Delta s_e = 0.5$, the scale change step can be derived using Equation (8).

$$\Delta s D = \Delta s_e + D \Rightarrow \Delta s = \frac{\Delta s_e}{D} + 1 \tag{8}$$

### 3.5. Geometric Correspondence

Template matching is a method that involves using one or more template images to search for matching patterns in other photos, depending on pre-existing patterns. The

shape matching method utilizes the shape characteristics of the template image to identify similarities in the shape characteristics of the target image, as depicted in Figure 5.

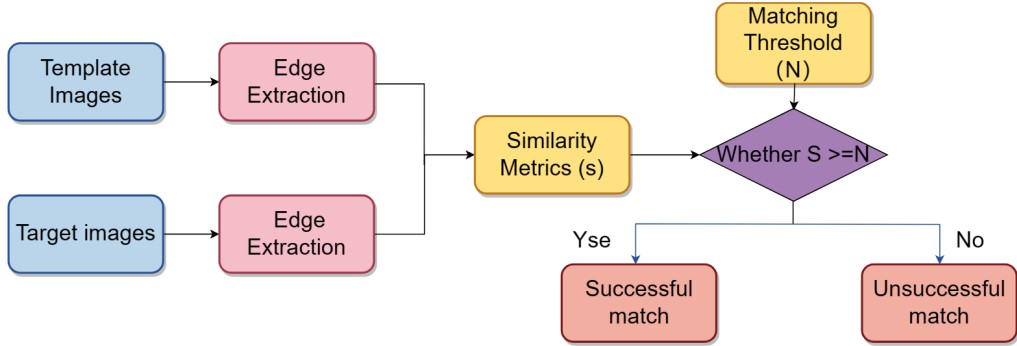

**Figure 5.** Geometric similarity assessment.

### 3.5.1. Feature Extraction

The contour features of the image are obtained by employing an edge detection algorithm [29]. Commonly employed edge detection operators are the Sobel operator, which is a first-order differential operator, the DoG operator, which is a second-order operator, and the Canny operator.

The Sobel operator approach is known for its computational simplicity and speed. On the other hand, the DoG operator is not only straightforward to implement and rapid in calculation, but it also demonstrates robustness to scale changes and rotation, resulting in the extraction of detailed local feature information. The Canny operator has a high rate of detecting edges and precise localization of the edge points. However, it is susceptible to noise and may result in the loss of finer features of the image's edges during the detection process. The edge detection results are shown in Figure 6.

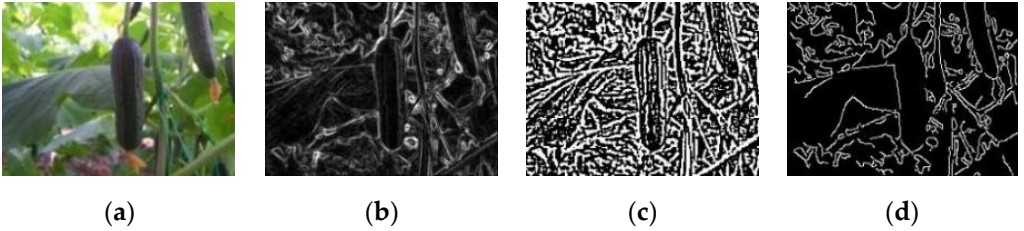

|       (a)       |       (b)       |       (c)       |       (d)       |

**Figure 6.** Edge detection: (**a**) the initial image; (**b**) Sobel edge detection; (**c**) difference of Gaussians (DoG); (**d**) Canny edge detection.

In comparison with the DoG operator and Canny operator, the Sobel operator detection yields detection outcomes that preserve the entire outline of the target and also demonstrate anisotropy. Hence, in this study, we opt for the Sobel operator for the purpose of detecting edges in images.

### 3.5.2. Metrics for Measuring Similarity

Shape matching involves the use of a similarity metric function to compute the similarity between the template picture and the image being compared [30,31]. Matching is deemed successful when the similarity surpasses a predetermined level.

The gradient vector of the desired edge points serves as the foundation for form matching. A point set is $m = \langle (x_i, y_i) | i = 1, 2, 3 \ldots n \rangle$ created using a template outline. The variable n is the number of points in the set, and the variable $d_i^m = (m_x, m_y)^T$ is the gradient vector associated with each point $m_x$ and $m_y$. The variables represent the gradient vectors in the directions $x$ and $y$, respectively. The matching point of the picture to be matched is obtained based on the positional relationship between the template reference

point and the edge point. Additionally, the gradient vector corresponding to the point $d_{x,y}^s = \left(s_{pi}(x,y), s_{qi}(x,y)\right)^T$ is also determined. During the matching process $w = (x,y)^T$, the similarity measure between the picture and the template is computed at a certain place. This computation is represented by Equation (9).

$$S(x,y) = \frac{1}{n}\sum_{i=1}^{n}\left(d_i^m, d_{x+x_i,y+y_i}^s\right) = \frac{1}{n}\sum_{i=1}^{n} s_{pi}s_q(x+x_i, y+y_i) \tag{9}$$

Nevertheless, the similarity metric function in Equation (9) is susceptible to changes in illumination. Therefore, it is imperative to normalize the equation, namely, through the mode-taking method for the edge gradient vectors, in order to reduce the impact of illumination.

$$S(x,y) = \frac{1}{n}\sum_{i=1}^{n} \frac{\left(d_i^m, d_{x+x_i,y+y_i}^s\right)}{\left\|d_i^m\right\| \cdot \left\|d_{x+x_i,y+y_i}^s\right\|} \tag{10}$$

According to the normalized correlation function in the above equation, its computational cost can be expressed as $O(kn)$, where k represents the number of template pixels and n represents the number of image pixels. The number of templates and the number of target image edge points determine the algorithm's efficiency. Given that the template picture is already established, we can enhance the effectiveness of template matching by decreasing the number of unnecessary edge points in the target image.

## 4. Experimental Results and Analysis

### 4.1. Laboratory Setting

The specific details of the experimental running environment can be found in Table 3.

**Table 3.** Laboratory setting.

| Parametric | Configure |
|---|---|
| Operating System | Windows 11 |
| CPU | 12th Gen Intel(R) Core(TM) i7-12700H 2.70 GHz |
| Image Processing Libraries | OpenCV, Halcon |
| Programming Language | C++, Python |
| IDE | Visual Studio 2019 |

### 4.2. Image Processing

4.2.1. Example Image Pre-Processing

The cucumber sample image undergoes several preprocessing stages, such as converting it to grayscale, using histogram equalization to enhance the image and improve its edge information, and binarizing it using the global thresholding OTSU method [32–34]. This approach iterates through all the different grayscale values in order to identify the ideal threshold, denoted as T = 126, that maximizes the interclass variance (ICV). Subsequently, the obtained threshold is utilized to partition the improved image, resulting in a binary image. Additional processing involves hole filling to eliminate noisy points on the image, as well as dilation and erosion processes to remove noisy points near the target edges. The image is subsequently reversed, as depicted in Figure 7.

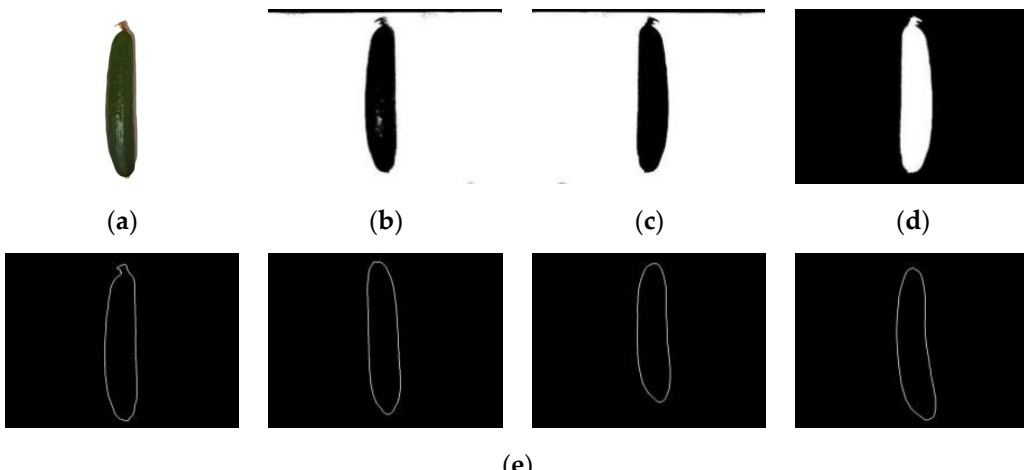

**Figure 7.** Preprocessing of images: (**a**) the initial image; (**b**) thresholding; (**c**) hole filling; (**d**) image mirroring; (**e**) edge extraction.

The inverted picture is employed to derive the target boundary by initially applying a morphological algorithm to dilate the image, followed by conducting a dissimilarity operation between the dilated result and the original target image. The boundary processing technique is applied to all cucumber sample photos to extract the boundary data. Several outcomes are depicted in Figure 7.

4.2.2. Target Image Preprocessing

The color balance of the cucumber target image is adjusted by performing contrast stretching on its different color components. The upper limit of each channel is defined as maxG ($\leq$255), and the LUT lookup table function is employed to convert the color gradient of a color channel from 0–255 to 0–maxG, thereby producing a reduction in color intensity of the color channel. Similarly, inside the HSV color space, the contrast of a particular channel can be expanded by utilizing the Look-Up Table (LUT) to modify the saturation, brightness, and other picture attributes.

We decrease the intensity of the green channel by 50% in the RGB color model and increase the saturation of the image in the HSV color model.

Following the color adjustment, the color contrast is intensified in the HSV color space. Consequently, the image with improved saturation is divided into segments inside the HSV color space. The H and S components are utilized for color distance representation, and distinct regions of varying colors are subjected to thresholding and blending for segmentation. The preliminary segmentation is set with a H range of 78 to 99 and a S range of 43 to 255 [35,36].

The image that was first segmented is converted into binary form and then subjected to morphological processing in order to eliminate smaller, unconnected sections. The processed image is superimposed onto the original image to get the segmented image, followed by the execution of edge extraction, as depicted in Figure 8.

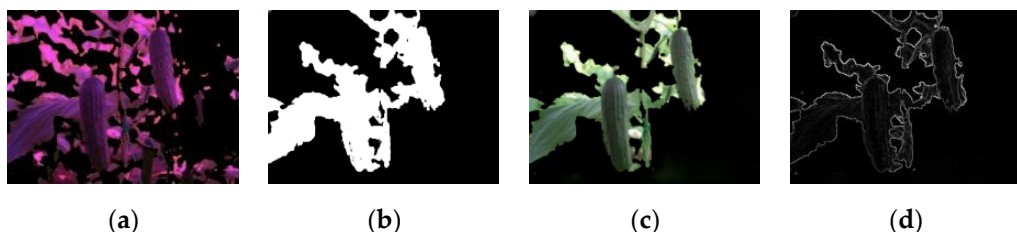

**Figure 8.** The target image preparation has four steps: (**a**) color segmentation, (**b**) binarized morphology, (**c**) image mask, (**d**) edge extraction.

After applying the Sobel operator for edge detection on the original image, there are 4939 edge points. Following color segmentation, the image contains 116 edge points. The template matching algorithm complexity is O(kn), resulting in 73 edge points in the template image. Using only one template reduces the algorithm's matching of the target image after color segmentation by 352,079 times, significantly enhancing the algorithm's efficiency.

### 4.3. Geometric Correspondence

#### 4.3.1. Image Placeholder

The image contours are reconstructed using the normalized Fourier descriptor, as depicted in Figure 9.

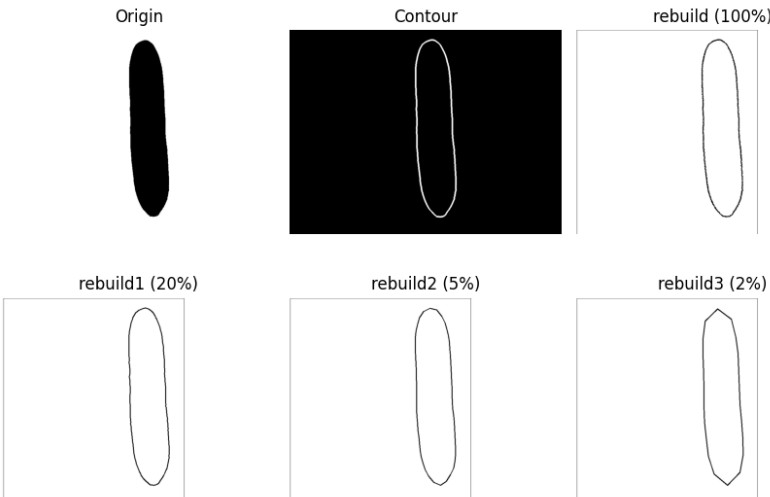

**Figure 9.** Reconstruction of edges using normalized Fourier descriptors.

The descriptor reconstruction with 100% fidelity is indistinguishable from the original boundary. The descriptor reconstructions with 5% and 20% fidelity preserve the primary characteristics of the boundary. However, the descriptor reconstruction with 2% fidelity lacks the primary features of the boundary and displays undesirable distortion. Hence, a minimal quantity of normalized Fourier descriptors is sufficient to define the boundary of the fruit.

The contours of the sample image were characterized using normalized Fourier descriptors. The top 5% descriptors were isolated and the associated descriptors of each individual image were averaged to reconstruct the contour, resulting in a standardized fruit contour template. The multi-template image is shown in Figure 10.

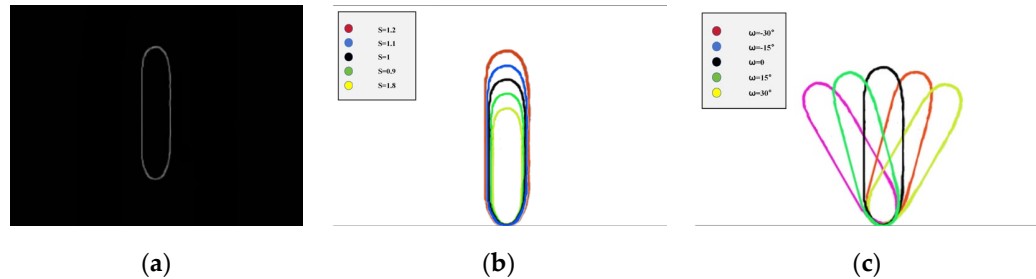

**Figure 10.** Edge of the template. (**a**) Template image, (**b**) Multi-angle templates (**c**) Multi-scale templates.

#### 4.3.2. Templates with Several Angles

The target cross diameter in the cucumber template image is approximately 25 pixels. Theoretical calculations indicate that the rotation step is 2.29°; however, practical experience shows that a step of 5° can also achieve accurate recognition. The rotation angle range for

cucumber fruit growth is fixed at $\pm 30°$, and the corresponding angle change coefficient table is provided in Table 4.

**Table 4.** Modification of template angle and scale.

| Serial No. | 1 | 2 | 3 | 4 | 5 |
|---|---|---|---|---|---|
| Angle Ratio | $-30°$ | $-15°$ | $0°$ | $15°$ | $30°$ |
| Scales Ratio | 0.8 | 0.9 | 1 | 1.1 | 1.2 |

### 4.3.3. Templates at Many Scales

Similarly, the scale step for the template image, which has a cross diameter of 25, is determined to be 0.1 based on precise calculations. The resulting scale change table is provided in Table 4.

### 4.3.4. Object Identification

The Sobel operator is used to extract edge information from the color-segmented image for matching purposes. Next, the dot product of the gradient vectors of the multi-scale template image and the image to be matched is computed using the similarity measure function.

The similarity between two points is determined by the inner product of their gradient vectors. If the inner product is 0, the similarity is also 0. Conversely, if the corresponding points match precisely, the inner product is 1. The magnitude of the inner product directly correlates with the level of similarity. Prior to the matching process, we set a predetermined threshold value. Once the inner product surpasses this threshold, we consider the matching to be successful. As seen in Figure 11, the spots that are correctly matched in the original image are shown by red rectangle boxes [37].

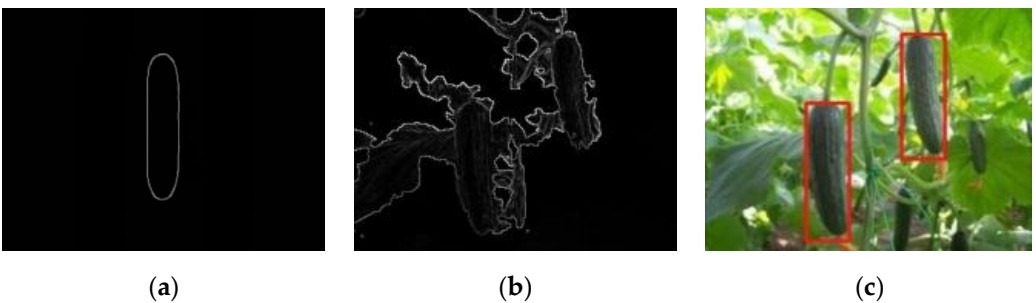

(**a**)  (**b**)  (**c**)

**Figure 11.** Identification of a certain target. (**a**) Template structure, (**b**) Edge data of the image to be compared, (**c**) Recognition outcomes.

### 4.4. Recognition of Targets at Several Scales and Angles

To determine the superiority of this approach, its findings were compared with those obtained by the standard unenhanced color segmentation recognition algorithm, template matching algorithm, and the SIFT and SURF feature point matching algorithms.

The comparison uses the G-component of the RGB model for the purpose of image segmentation, subsequently applying the process of fitting rectangular shapes. The form matching algorithm utilizes the template given in this work, employing an equivalent number of templates. The OpenCV library employs the SIFT and SURF algorithms for feature matching. The results of multi-scale and multi-angle recognition are shown in Figure 12.

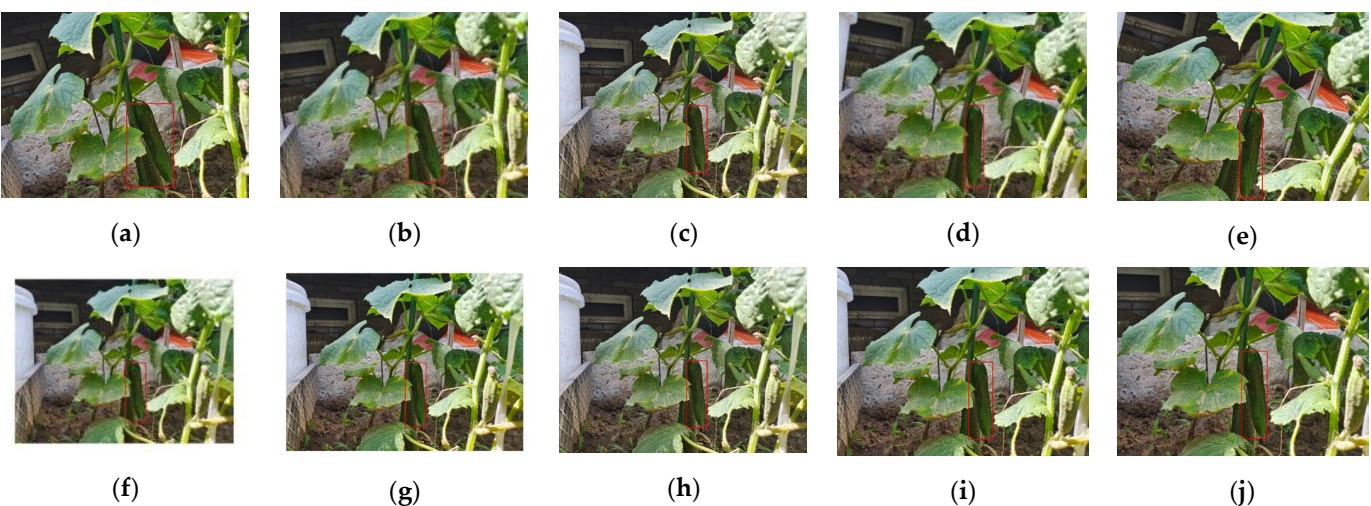

**Figure 12.** Multi-angle and multi-scale cucumber fruit identification: (**a**) ω = −15°; (**b**) ω = −10°; (**c**) ω = 0°; (**d**) ω = 10°; (**e**) ω = 15; (**f**) S = 0.8; (**g**) S = 0.9; (**h**) S = 0.1; (**i**) S = 1.1; (**j**) S = 1.2.

The results obtained from various algorithms are displayed in Table 5.

**Table 5.** Results of various algorithms for multi-angle and multi-scale recognition.

| Algorithm | ω = −15°/S = 0.8 | ω = −10°/S = 0.9 | ω = 0°/S = 1 | ω = 10°/S = 1.1 | ω = 15°/S = 1.2 |
|---|---|---|---|---|---|
| RGB_Color | ×/× | ×/× | ×/× | ×/× | ×/× |
| Shape Match | √/√ | √/√ | √/√ | √/√ | √/√ |
| SIFT | ×/√ | √/× | √/√ | √/√ | √/× |
| SURF | ×/× | ×/√ | √/√ | ×/× | ×/× |
| Color and Shape | √/√ | √/√ | √/√ | √/√ | √/√ |

ω is the angle change factor; S is the scale change factor.

The table illustrates that the form matching algorithm successfully matches at different angles and scales, providing evidence that the system can precisely identify the target within a specific range of rotation angles and scales. The color segmentation method is invariant to changes in target scale and is exclusively driven by lighting conditions. However, SIFT and SURF algorithms do not yield a high recognition rate when applied to multi-scale targets.

For this experiment, we have selected photographs of cucumbers that are developing at different sizes and orientations. These photographs exhibit diverse levels of light intensity, and the subject is somewhat obscured. The results of the recognition are shown in Figure 13.

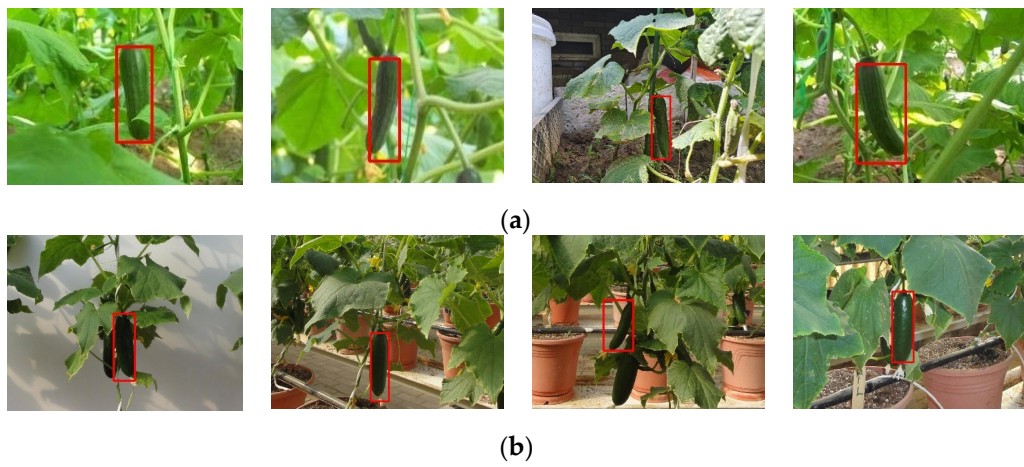

**Figure 13.** Target recognition: (**a**) greenhouse shooting images; (**b**) database images.

### 4.5. Identification of Fruit Fields

In order to evaluate the effectiveness of the algorithm proposed in this paper, we employed an industrial camera integrated with this algorithm to perform real-time experiments for cucumber fruit recognition in greenhouse environments.

The trial was conducted in XiangYou Vegetable Professional Cooperative in Jingyang County, Xianyang City, Shaanxi Province. A light barrier was readied before the experiment to prevent overexposure in the presence of intense light. The camera was configured to move horizontally around 40 cm in the target region according to the focal length of the lens and the target size.

The industrial camera was linked to the hardware device via USB. HALCON was used to establish the connection, open the camera, and choose the real-time image. A template was then generated in advance, with the target image being the real-time image captured by the camera. Parameters were configured as follows: pyramid level 5, template scale change range of $\pm 0.2$, angle range of $\pm 15°$, step sizes of 0.1 and $5°$, and a greed factor of 0.7. The target was identified in the real-time image using a green line and enclosed within a red rectangle box. The specific recognition results are shown in Figure 14.

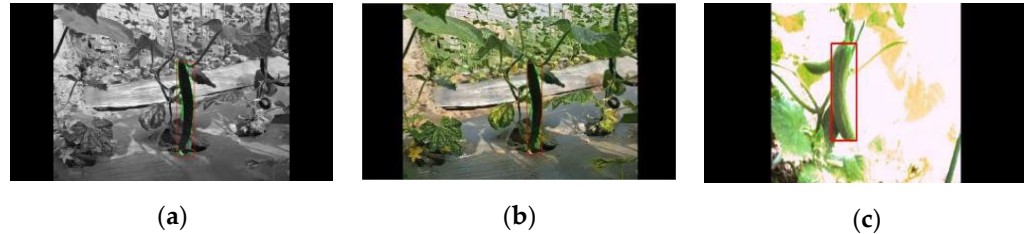

|  (**a**)  |  (**b**)  |  (**c**)  |

**Figure 14.** Cucumber target field identification: (**a**) gray; (**b**) RGB; (**c**) exposure.

During the cucumber real-time recognition trials, the color space for gray, BGR24, and image exposure were acquired in real time. The mean recognition time was 38.9 frames per second (FPS), with the lowest matching score recorded as 0.8. The image depicts the real-time segmentation effect and the impact of selecting the maximum outer rectangle box. It is evident that both the gray image and the BGR image successfully identify the fruit target. Additionally, the image showcases the real-time recognition of the cucumber. Although the real-time segmentation effect is somewhat absent in the experiment, the final fruit is effectively identified.

Overall, the algorithm used in this study has a significant positive effect on the recognition of cucumber fruit and can be applied in practical settings.

## 5. Discussion

Existing agricultural machinery faces challenges in performing intricate and precise tasks. This has led to a growing need for agricultural robots that possess advanced autonomous capabilities, including perceptual decision-making and eye-hand coordination. The crop growth conditions necessitate sensing technology that is both very precise and immediate, as well as cost-effective and straightforward in terms of algorithmic design. The template matching technique is better suitable for agricultural machinery and equipment compared with machine learning-based recognition algorithms due to its simplicity and ability to handle various items without requiring extensive training costs. This study employs this approach, in conjunction with other target recognition methods, to accurately detect cucumber targets. Recognition investigations are conducted on target photographs collected in greenhouse settings with different lighting conditions, and the results of the recognition are presented in Table 6.

**Table 6.** Cucumber target recognition results of several algorithms.

| Algorithm | Correctly | Incorrect | Accuracy | Average Time/ms |
|---|---|---|---|---|
| RGB_Color | 120 | 80 | 60% | 26.83 |
| Shape Match | 168 | 32 | 84% | 198.21 |
| SIFT | 92 | 108 | 46% | 118.61 |
| SURF | 86 | 114 | 43% | 91.42 |
| Color and Shape | 172 | 28 | 86% | 35.71 |

The aforementioned experimental findings demonstrate that the algorithm suggested in this research surpasses other conventional algorithms in terms of recognition rate and matching efficiency. The color segmentation algorithm faces difficulties in effectively identifying the cucumber target in its natural context due to the presence of a background that closely matches its color. The SIFT and SURF features are considered local features, meaning they only capture information from a small region of an image. As a result, they are not suitable for accurately representing the overall characteristics of a target in a specific scene and, hence, cannot be effectively used for target matching. The conventional form matching algorithm has a commendable identification rate, albeit with an accompanying increase in matching time as the number of templates grows. The technique described in this study aims to decrease the time required for matching by first dividing the color information, which, in turn, greatly reduces the amount of edge information in the image that needs to be matched.

## 6. Conclusions

This work focuses on the difficulty of precisely and efficiently recognizing targets for agricultural harvesting robots. This challenge arises mainly from the presence of backgrounds with similar colors, complicated growing settings, and irregular lighting conditions, all of which make it harder to separate fruit targets from their surroundings. A novel target recognition method is developed to precisely identify fruit targets, utilizing color and shape information. The research subject chosen for this study is cucumber. The method exploits the distinctive form characteristics of the fruit, which are markedly dissimilar to those of the stem and leaves. The study presents a multitude of fruit target picture templates and outlines the process of reconstructing their borders to produce the standard template contour using the normalized Fourier descriptor. Additionally, in order to accommodate the nonuniform growth of cucumber, the scale and rotation angle are computed to enhance the precision of matching. Integrating multi-scale template matching results in an abundance of template data, hence causing an increase in the time required for matching. In order to enhance the accuracy of real-time matching, the image undergoes preprocessing using an algorithm that segments colors based on the HSV color space. This procedure effectively reduces the area to be matched, resulting in improved operational efficiency of the method.

Recognition studies are conducted using 200 photos obtained from the field, and are compared with four classic recognition methods. The method presented in this research demonstrates high accuracy and efficiency in target recognition, particularly in the context of identifying targets within greenhouse environments. The method is straightforward, the necessary hardware resources are cost-effective and adaptable, and it may be effectively utilized in agricultural robots, making a substantial contribution to the field of agricultural robot vision.

Future research should prioritize enhancing the algorithm by transferring and rectifying it on hardware devices, as well as expanding its applicability to the practical domain of fruit recognition.

**Author Contributions:** Investigation, W.L.; methodology, H.S.; software, H.S.; supervision, W.L.; writing—original draft preparation, H.S.; writing—review and editing, W.L., Y.X. and J.K. All authors have read and agreed to the published version of the manuscript.

**Funding:** This work was supported by the Youth Fund of the National Natural Science Foundation of China (Grant No. 62203285) and the Youth Fund of Shaanxi Provincial Natural Science Basic Research Program General Project (Grant No. 2022JQ-181).

**Institutional Review Board Statement:** Not applicable.

**Informed Consent Statement:** Not applicable.

**Data Availability Statement:** The dataset can be found at: https://www.kaggle.com/datasets/farahseifeld/greenhouse-cucumber-growth-stages (accessed on 21 February 2024).

**Acknowledgments:** The authors would like to thank the anonymous reviewers for their constructive comments and suggestions.

**Conflicts of Interest:** The authors declare no conflicts of interest.

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
