# Peer review of "Real-Time Cucumber Target Recognition in Greenhouse Environments Using Color Segmentation and Shape Matching"

_applsci, doi:10.3390/app14051884_

Round 1

Reviewer 1 Report

Comments and Suggestions for Authors

Good job! Please check my comments below.

1.      You mentioned “Vitzrabin[5] achieved the initial separation of red bell peppers by creating a 3D Normalized Difference Index (NDI) space and establishing three predetermined thresholds.”. Is it 3-dimensional instead of 2-dimensional study? You talked about color in this paragraph, is this also used color in 3D?

2.      You missed the citation of this sentence: “Li substituted the green component in the RGB color space with the hue component (H2) …”

3.      I was kind of lost in section 3, what is the purpose for each step? What is the final output for the model to do cucumber recognition?

4.      There are so many figures and tables, please try to combine some or move to supplementary material.

5.      Did you try YOLO algorithms? How’s the performance comparison between yours and YOLO?

Author Response

We appreciate your professional review work on our articles!We have responded to your questions and amendments line by line, as reflected in the document.

Reviewer 2 Report

Comments and Suggestions for Authors

The algorithm utilizes Fourier descriptors to describe and reconstruct the edge information of the cucumber sample, generating a matching template image. Color conditioning in the HSV color space enhances the segmentation of the target region, improving the effectiveness of shape matching.

It is important to mention the significance of the cucumber and its scale, but with statistics and references.

The paper does not discuss the computational complexity or real-time performance of the algorithm, which could be important considerations for practical implementation.

In my opinion, this type of work should make the data used available, in this case, the images.

The proposed methodology, the need for a white background, its operation in a real environment, and its implementation in a portable system should be better explained and connected.

In my opinion, the work lacks depth and clarity. The arguments are not sufficiently justified, and it is not clear to me why these techniques are used instead of more modern ones like CNNs for segmentation, at least.

Author Response

(The authors gave the same response as above.)

Round 2

Reviewer 1 Report

Comments and Suggestions for Authors

Looks great to me!

Happy New Year!

Reviewer 2 Report

Comments and Suggestions for Authors

The reviewer appreciates the authors' rigorous and meticulous efforts in addressing the feedback provided. These revisions have significantly enhanced the clarity and impact of the work, making it a valuable contribution to the engineering community.